# Obtaining free USArray data by multi-dimensional seismic reconstruction

Yangkang Chen[1]*, Min Bai [2] & Yunfeng Chen[3]

USArray, a pioneering project for the dense acquisition of earthquake data, provides a semi-uniform sampling of the seismic wavefield beneath its footprint and greatly advances the understanding of the structure and dynamics of Earth. Despite continuing efforts in improving the acquisition design, network irregularity still causes spatial sampling alias and incomplete, noisy data, which imposes major challenges in array-based data analysis and seismic imaging. Here we employ an iterative rank-reduction method to simultaneously reconstruct the missing traces and suppress noise, i.e., obtaining free USArray recordings as well as enhancing the existing data. This method exploits the spatial coherency of three-dimensional data and recovers the missing elements via the principal components of the incomplete data. We examine its merits using simulated and real teleseismic earthquake recordings. The reconstructed P wavefield enhances the spatial coherency and accuracy of tomographic travel time measurements, which demonstrates great potential to benefit seismic investigations based on array techniques.

[1] School of Earth Sciences, Zhejiang University, Hangzhou, Zhejiang Province 310027, China. [2] Key Laboratory of Exploration Technology for Oil and Gas Resources of Ministry of Education, Yangtze University, Wuhan, Hubei Province 430100, China. [3] Deep Earth Imaging, Future Science Platform, CSIRO, Perth, WA 6151, Australia. *email: chenyk2016@gmail.com

The past two decades have witnessed a major proliferation of broadband seismic arrays around the globe. One excellent example is USAarry, the seismology component of the national Earth science program EarthScope, that migrated continuously across the North American continent (year 2004–2013; now deployed in Alaska and northwestern Canada), providing a dense sampling of the seismic wavefield beneath its footprint. The availability of a large amount of high-quality data offers unique opportunities for high-resolution seismic imaging, which leads to a much-improved understanding of the Earth's structure at various scales[1–11]. The relatively even-spaced sensors, with an average spacing of ~70 km, form an ideal network configuration for the application of array-based methods[12,13]. While the quantity of array data is continuously growing, the improvement of data quality is hampered by the presence of noise and a biased spatial sampling that results from unevenly distributed earthquake sources and stations. Both factors (i.e., quantity and quality) play an important role in exploiting the data and achieving robust imaging outputs. Most data improvement efforts have been focusing on increasing the data volume whereas generally less attention was paid to improve the quality of existing recordings. The high-quality data with a regular spatial sampling can greatly benefit data analysis, processing and visualization, and also improve the numerical stability and accuracy for grid-based seismic imaging techniques (e.g., Eikonal[4] and Helmholtz tomography[14]). As a result, regularization approaches that suppress the noise and interpolate the missing traces of array data are highly demanded[15–17].

Some effective methods for seismic data reconstruction have been proposed in the exploration seismology community. One type of the most widely used methods is based on a sparse transform that maps the seismic signals to certain domains (e.g., Fourier[18], curvelet[19], slant stacklet[20], and seislet domains[21]), where the useful information can be sparsely represented and separated from the missing data and random noise. Another type of mainstream methods is based on the Cadzow filtering or the singular spectrum analysis (SSA)[22], which is a rank-reduction-based method that transforms the data to frequency-wavenumber or frequency-space domains to extract the spatial coherency of the entire dataset for reconstructing the missing information. Oropeza and Sacchi[23] extended the SSA method to multi-channel version to tackle the 3-D seismic data reconstruction challenge and later on Kreimer et al.[24] formulated the high-dimensional reconstruction as a nuclear-norm constrained tensor completion problem in the frequency-space domain. More recently, Chen et al.[25] improved the traditional truncated singular value decomposition (TSVD) method by deriving a new rank-reduction formula that better decomposes the data space into signal and noise subspaces. All aforementioned reconstruction methods are commonly applied to seismic data from exploration-scale surveys, while their applications to earthquake data, especially the multi-dimensional data recorded by large-scale seismic arrays, have been seldom reported.

In global seismology community, several methods have been proposed to interpolate the irregularly sampled seismic data. Most earlier studies have been concentrating on interpolating the receiver functions[26] that consist of P to S converted waves based on a certain form of spatial smoothing, including, for example, weighted stacking with either linear[27], gaussian[28,29], or cubic spline functions[30,31]. Also implemented are methods based on high-resolution Radon transform[32], singular spectrum analysis[33,34] and, more recently, non-linear waveform stretching-and-squeezing[35]. Aside from interpolating the receiver functions, Schneider et al.[17] reconstructed the weak-amplitude under-side reflections from the mantle transition zone (i.e., PP precursors) while utilizing a compressive-sensing-based approach that seeks the sparsity of dominating energy in the frequency-wavenumber domain. A recent study also applied the idea of compressive sensing to reconstruct the synthetic surface wavefields via a sparse representation using a plane-wave basis[36]. These earlier studies represent continuing efforts of the global seismology community in improving the earthquake data toward a regularly sampled wavefield. However, challenge still exists in view that (1) only a specific type of data (e.g., receiver functions, PP precursors), which is typically structurally simple, is reconstructed and (2) the energy of useful signals tends to be excessively smoothed by the ad hoc interpolation schemes, which limits the data resolution thereby the resolvability of small-scale structures.

In this study, we develop an effective framework to reconstruct the three-dimensional (3D) data of an earthquake recorded by USArray. We propose a localized rank-reduction method to simultaneously reconstruct the missing traces and improve the weak-amplitude phase arrivals. Compared with the regular global rank-reduction method, our localized approach is superior at preserving the small-scale features in the array data, which is critical for high-resolution imaging of subsurface structures. We demonstrate its signal-improvement capability via a synthetic dataset and then apply the proposed method to January 18, 2009, Kermadec Islands Mw 6.4 earthquake recorded by USArray, where a significant portion of the recordings is missing due to deployment limitations. The reconstructed earthquake wavefield provides the virtual recordings at the missing locations as if they had been acquired by the actual stations during the earthquake. We demonstrate the merits of the proposed method and data improvement by conducting cross-correlation measurements of P-wave arrival times, a fundamental step in body-wave travel time tomography.

## Results

**Synthetic test**. We first conduct a synthetic test to demonstrate the performance of the proposed methodology. The model is designed to honor the complex Moho structure in central US[37] that is characterized by large depth variations by as much as 20 km (Fig. 1). The elastic properties (velocity and density) of the crust and upper mantle are obtained from the average values of AK135 continental model[38] (Fig. 1a–c). We synthesize the recorded wavefield by solving the 3D elastic wave equation based on a finite-difference method. (A different synthetic example test is provided in the Supplementary Note 1 and Supplementary Figs. 1–6). We simulate plane-wave incidence of teleseismic earthquake by simultaneously injecting energy from multiple point sources located on a plane in the upper mantle (Fig. 1d). The physical parameters of an earthquake (e.g., epicenter and magnitude) are implicitly considered in our simulation. The direction of wavefront is determined from the epicenter distance and azimuth of the earthquake investigated in this study. The stacked P-wave of actual earthquake recordings is employed as the effective source-time function to honor the actual earthquake source parameters and ensure a similar frequency content and signal energy of the synthetics.

The 3D seismic data consist of 200 time samples with a sampling rate of 0.5 s and 100 equally spaced spatial samples in both longitudinal (X) and latitudinal (Y) directions (Fig. 2a). The vertical component of the simulated wavefield shows clear P and Moho converted phases (Ps) (Fig. 2a). To mimic the realistic signal-to-noise ratio (SNR), we add real noise that precedes the first arrivals (i.e., P waves) from the USArray data to the synthetics (Fig. 2b). The missing traces are generated by applying the sampling matrix (Fig. 2c) to the noisy data. The final simulated data contain 30% missing traces (Fig. 2d). To quantitatively evaluate the data quality, we use a SNR criterion

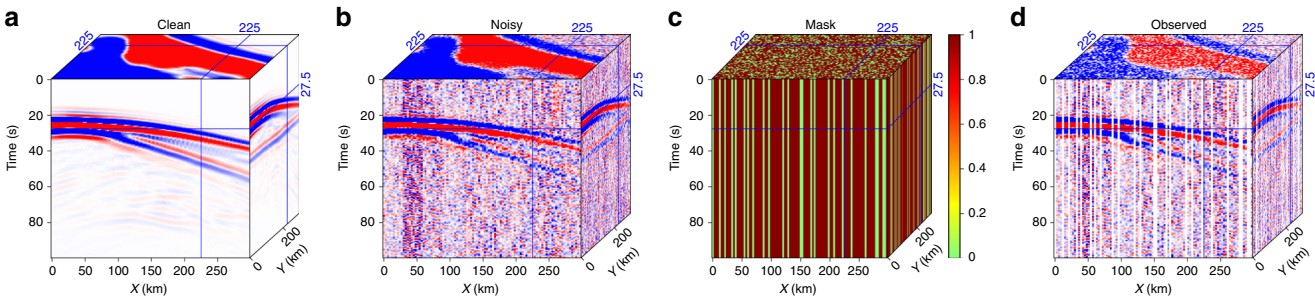

**Fig. 1** Wavefield simulation of plane-wave incidence beneath a regional array. The synthetic model is constructed using real Moho constraints[37] and elastic parameters of **a** P-wave velocities, **b** S-wave velocities, and **c** densities from a 1D reference model[38]. The blue lines mark the location of horizontal and vertical slices, where the values of model parameters are demonstrated and projected onto the corresponding sides of the model cube. **d** Observation system demonstrating the positions of planar source and receivers. The stars at the surface denote the evenly distributed receivers and symbols on the dipping plane denote the injected source positions

**Fig. 2** The vertical component of the synthesized data. **a** The clean data obtained from elastic wave simulation. **b** The noisy data with signal-to-noise ratio equal to 10.12 dB. **c** The sampling matrix with a sampling ratio of 70%. **d** The final data (signal-to-noise ratio equals to 4.35 dB) after applying the sampling matrix to noisy data in (**b**)

defined as follows: $\mathrm{SNR} = 10\log_{10}\frac{\|\mathbf{s}\|_2^2}{\|\mathbf{s}-\hat{\mathbf{s}}\|_2^2}$, where $\mathbf{s}$ denotes the exact solution (i.e., the clean data), and $\hat{\mathbf{s}}$ is the noise contaminated or incomplete data. The respective data quality metrics (SNR) for the noisy and incomplete data are 10.12 and 4.35 dB, suggesting a significantly decreased data quality caused by missing traces. This criterion is also applied to assess the quality of reconstruction, where $\hat{\mathbf{s}}$ represents the recovered data using the global or localized reconstruction algorithms. Thus, the SNR measures the deviation of an estimated data from its true solution.

The global rank-reduction method assumes that the seismic data are composed of several plane waves, however, this assumption is often violated due to the presence of non-planar wavefields in realistic models (see Fig. 1). Thus, we improve the global rank-reduction method using a localized scheme. This method divides the data cube into several smaller volumes to minimize the curvature of seismic waves, which essentially imposes a local plane-wave constraint to alleviate the non-planar effects. More sophisticated methods based on non-plane-wave assumptions (e.g., non-stationary principal component

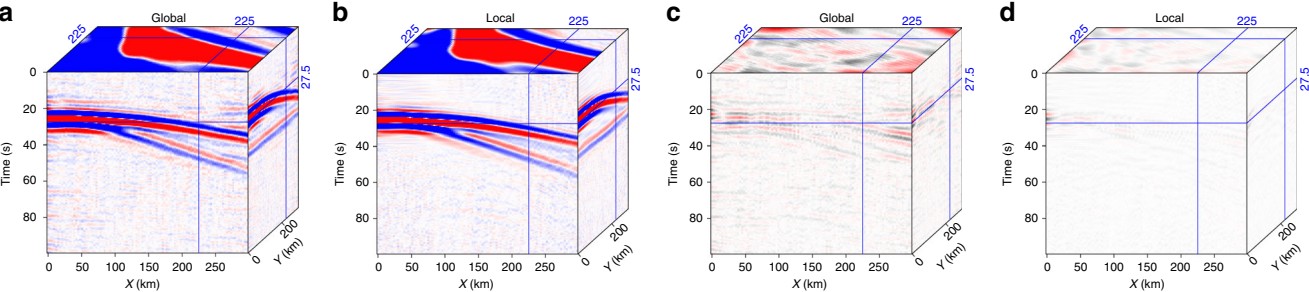

**Fig. 3** Synthetic test results. **a** Reconstructed data using the global rank-reduction method. The signal-to-noise ratio of the reconstructed data is 17.34 dB. **b** Reconstructed data using the localized rank-reduction method (signal-to-noise ratio is 21.53 dB). **c**, **d** Reconstruction errors corresponding to (**a**, **b**)

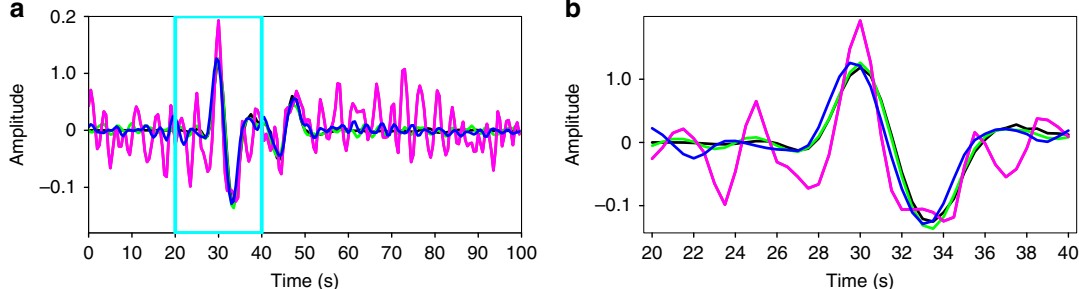

**Fig. 4** Comparison of the single trace amplitude ($X = 225$, $Y = 225$) in different scales. **a** Comparison in the original scale. **b** Comparison in the zoomed-in scale. The black and pink lines represent clean and noisy data, respectively. The trace processed using global and localized rank-reduction methods are shown by the blue and green lines, respectively. The black line (exact solution) is almost completely overlapped with the green line (localized reconstruction), indicating that the error of reconstruction is minimal for the proposed approach

analysis[39]) may offer better solutions to the curved wavefronts. To implement the localized rank-reduction method, we apply a 3D moving window with a size of $50 \times 50 \times 50$ to the data volume. We allow a 25-points (i.e., 50%) overlap in each direction between successive windows to suppress the edge effects. For each subset of the data volume within the moving window, we use a reference rank of three to compute the trajectory matrix (see Methods), whereas a reference value of nine is adopted in global rank-reduction method. In the global case, the two plane waves are well reconstructed (Fig. 3a), but the curved wavefront exhibits significant reconstruction errors, particularly in areas with large curvatures (Fig. 3c). In comparison, the localized rank-reduction method does not cause obvious damages to the signal (Fig. 3b) and the reconstruction error is almost zero everywhere (Fig. 3d). Based on SNR metric, the respective quality metrics for the global and localized reconstructions are 17.34 and 21.53 dB, suggesting a better performance of the latter approach. We extract a single trace at a location where large curvature exists (Fig. 4). The reconstructed trace using localized rank-reduction method (green) almost fully recovers the input (black; Fig. 4b), whereas an obvious time shift is present in the trace obtained from the global method (blue; see Fig. 4b).

We also use local similarity[40,41], which measures the similarity between two signals in a local sense, to evaluate the reconstruction performance. A mathematical introduction of the local similarity metric is provided in Supplementary Note 2. In our case, we intend to obtain a reconstruction result that is as close to the clean data as possible. The local similarity between the clean and the reconstructed data using the global rank-reduction method shows obvious low values along the curved wavefront (Fig. 5a). In comparison, the local similarity of localized reconstruction is high throughout the data volume regardless of the shape of the wavefront (Fig. 5b). We further investigate the effects of sampling ratios (i.e., the percentage of missing data) on the reconstruction. To this end, we vary the sampling ratios from

90 to 10% and randomly remove the traces. The test results show that (1) the reconstruction performance, which is defined as SNR using Eq. (8), improves with an increasing sampling ratio, (2) the proposed algorithm is robust even at the low end (30–40%) of the sampling range, and (3) the localized rank-reduction implementation always achieves superior reconstruction performance than the global method (Fig. 6).

The parameters for both strategies are fine-adjusted to achieve the best reconstruction results. The only parameter for the global rank-reduction method is the rank. To determine the optimal value, we linearly increase the rank and select the one that maximizes the SNR of the reconstructed data. On the other hand, a two-step process is adopted to determine a pair of parameters (i.e., rank and window size) for the localized rank-reduction method. As a first step, we optimize the window size while considering a relatively large reference rank (e.g., five). The exact choice of reference rank is not critical at this stage since the rank selection will be further optimized by the automatic rank selection process. We fix the overlap between two neighboring windows to half of the window size. The length of window in each dimension needs to be determined properly such that the data are segmented into patches with the smallest extension. For example, for a dimension of 100, the recommended window sizes are 10, 20, 50, or 100, since other choices will all cause an extension of the dimension. With all possible combinations of the window sizes considering all three dimensions, we select the best window size leading to the largest SNR. In the second step, we further optimize the reference rank for the selected window size. We decrease the reference rank and adopt the new value if SNR can be further improved. In real data processing, we use the same strategy except for the criterion to evaluate the output performance, which is prohibited by a lack of ground-truth solution (i.e., term **s** in the equation of SNR). Instead, we define the maximum cross-correlation value between a reconstructed missing trace and its nearest observed trace as the criterion.

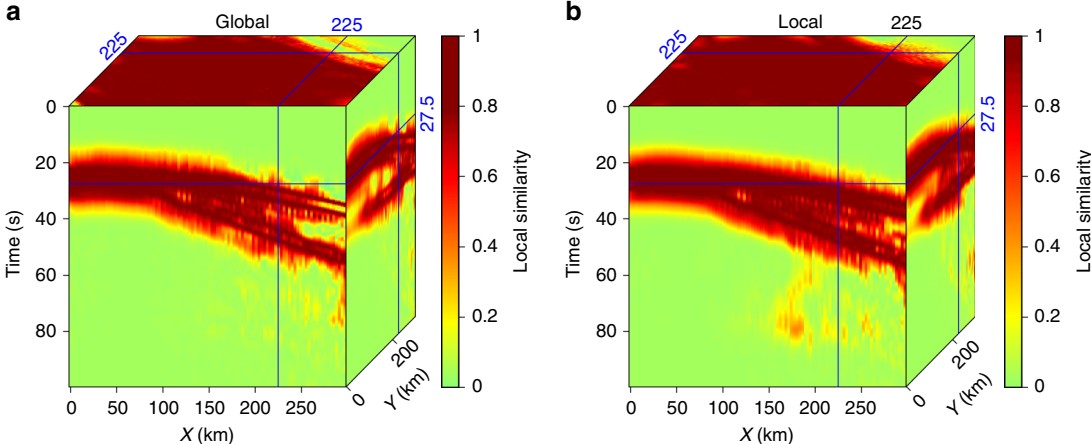

**Fig. 5** A comparison of reconstruction performance in terms of local similarity metric. **a** Local similarity using the global rank-reduction method. **b** Local similarity using the localized rank-reduction method. Note that local similarity around curved wavefronts is noticeably lower in the global reconstruction results

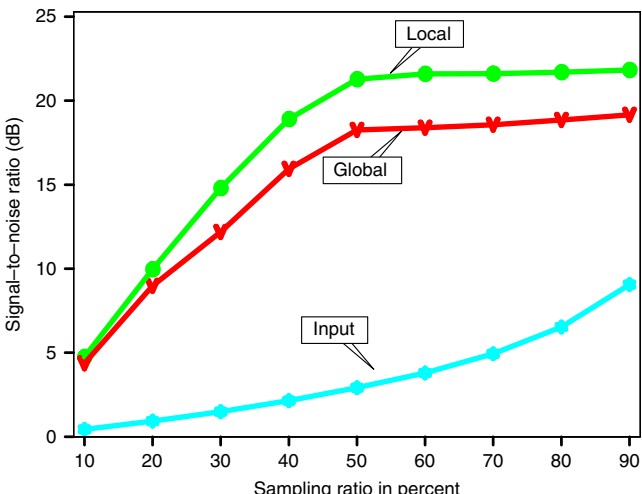

**Fig. 6** Reconstruction results at different sampling ratios. The reconstruction performance (signal-to-noise ratio) improves with the increasing sampling ratio. The localized rank-reduction method always outperforms the global algorithm

To investigate the influence of frequency content to the reconstruction performance of the presented methodology, we extract different frequency contents of the incomplete synthetic data for the reconstruction. We linearly increase the frequency band from 0.05 to 0.4 Hz at an interval of 0.05 Hz and perform reconstruction at each frequency slice (Supplementary Fig. 7). The seismic signals are well recovered in the reconstructed data for frequencies up to 0.3 Hz, showing more coherent P and Ps arrivals. The reconstruction performance degrades at higher frequencies (>0.3 Hz) with a lower degree of recovery of the missing traces. The performance of reconstruction is mainly limited by the weak high-frequency seismic signals with energy that is close to (or below) the noise level (see Supplementary Fig. 7g, h). We further perform a quantitative assessment of the reconstruction quality by computing the SNR of the output signal at each frequency (Supplementary Fig. 8). Compared with the raw data, the reconstruction improves the SNRs by a factor of two at all frequencies except for the high end (>0.3 Hz) of the frequency spectrum, where the SNRs fall below the level of input values. In summary, our test results suggest that the frequency content of the signal largely controls the performance of reconstruction, and high-quality results are achievable within the dominant frequency band (0–0.3 Hz) of the signal (Supplementary Fig. 9). As a result, a careful frequency analysis of the data is recommended prior to applying the reconstruction algorithm.

**USArray data test**. A more challenging test is performed using the real data from January 18, 2009, Kermadec Islands, New Zealand, Mw 6.4 earthquake recorded by the Transportable Array (TA) component of USArray (Fig. 7). Despite the best effort in acquisition design, earthquake data are rarely recorded on a perfectly even-spaced seismic array. Similar to earlier interpolation methods[17,28,34], the first step of our reconstruction algorithm is to bin the data onto a regular grid. To this end, we use a weighted interpolation method for the binning process. At each node location, the waveforms from nearby stations that are located within one grid distance are stacked and subsequently assigned to the node, whereas the node remains empty if no neighboring stations are available. We use a grid with respective dimensions of 1.0° and 0.5° in latitude and longitude directions, which is comparable to the station spacing of USArray, to minimize the effect of spatial smoothing while preserving small-scale features. Future study would focus on interpolating an arbitrary geometry given a randomly distributed dataset. The degree of data completeness after the binning process is demonstrated by the sampling matrix (Supplementary Fig. 10), where empty and data-filled grid points are indicated by zero and one, respectively. We obtain a regular grid with a dimension of 16 × 28 (longitude × latitude) that comprises a total of 448 sampling points. Among them 253 nodes are filled with the recorded earthquake data while the remaining 164 points require to be reconstructed by the proposed algorithm, resulting in a sampling rate (i.e., data completeness) of 56.5% in the data.

During the earthquake, TA was deployed in western US covering the Basin and Range Province and the Great Plains, which marks a transition region from the active western US to the relatively stable eastern part. This diverse tectonic environment provides an ideal dataset to test the robustness and accuracy of our reconstruction algorithm. We aim to create a complete data volume using the vertical component seismograms acquired at 448 stations. The same method can be applied to reconstruct the other two horizontal components. As a first step, the original traces are subject to a sequence of preprocessing including instrument response removal, integration to displacement, and binning onto a regular grid (Supplementary Note 3). The

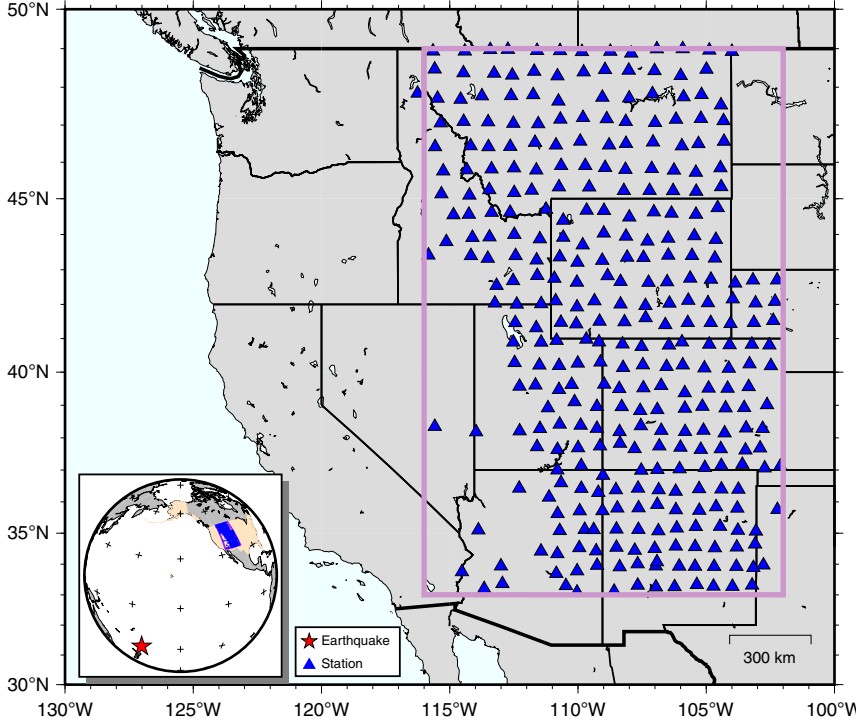

**Fig. 7** The distribution of USArray stations in this study. The lateral dimension of the reconstructed data volume is highlighted by the rectangle. A large portion of the data is missing due to relatively spare station coverage in the southwest. The inset shows the location of the earthquake (red star) relative to the recording array (blue triangles)

processed data forms a 3D cube with a dimension of 5400 × 16 ×28 along time, longitude and latitude axes (Fig. 8a). The quality of this dataset is mainly limited by (1) the missing traces that account for a significant portion of the data volume and (2) strong noises that interfere with phase arrivals. The former constraining factor leads to reduced resolution in regions with incomplete data sampling (e.g., the circled area on Fig. 8a) and the presence of noise can mask relatively low-amplitude arrivals such as the P waves (see Fig. 8a).

We apply the two rank-reduction methods to the preprocessed data. Figure 8b shows the reconstructed data using the global rank-reduction method. In global reconstruction, we treat the whole 3D data cube as the input and set the rank to eight, which is determined carefully by our tests. The reconstruction results show an improved data coherency compared with the input, as demonstrated by a more detailed wavefield variation in the time slice at 2600 s (Fig. 8a, b). However, a major issue with the global method is that it tends to smooth small-scale arrivals (e.g., P waves indicated by red arrows in Fig. 8b). In this example, because of the relatively weak energy of the first arrivals, they cannot be robustly distinguished from the noise using the global rank-reduction algorithm. This issue is largely resolved by the localized rank-reduction method, where both time slices and the first arrivals are well recovered (Fig. 8c).

We compare two time slices that contain weak phase arrivals between the original and the reconstructed data. The first time slice shows the energy related to PP phase (Fig. 9a, b), the free surface multiple of P wave. This weak phase is severely contaminated by the scattered energy associated with the P-wave codas. As a result, the wavefield pattern of PP is largely incoherent across the recording array, even in the center and northeastern portions where station coverage is high. The second time slice focuses on the wavefield around the excepted arrival time of SS phase, which is the shear wave reflected off the surface at the midpoint of source-station path, showing much coherent

energy compared to the scattered wavefield (Fig. 9c, d). The reconstructed time slice successfully fills the data gap and captures the detailed variation in wavefield energy. We demonstrate the reconstruction performance on full seismogram in Fig. 10, where nine traces are missing in this longitude slice. The quality of various phase arrivals is severely degraded by noise. After reconstruction, these phases are clearly identifiable from both pre-existing and reconstructed traces. A more detailed examination of weak-amplitude phases shows that (1) the waveform characteristics (e.g., phase and amplitude) of the existing traces are well preserved by the reconstruction algorithm without excessive smoothing and (2) the reconstructed traces well capture the coherent energy of the data, showing similar waveform quality to the nearby (observed) traces (Fig. 10c, d).

We conduct a bootstrapping test[42] to estimate the effect of spatial sampling of wavefield on reconstruction. We randomly select 40% of the observed seismograms to reconstruct the 3D data cube and repeat this step 20 times. We calculate the standard deviation of the 20 reconstructed datasets and use the normalized deviation as an estimate of uncertainty. The reconstruction uncertainty is low for body-wave phases (e.g., P and S) even in the presence of intermediate (200 km) data gap (Fig. 10e), whereas the uncertainty is slightly higher in time ranges with weak arrivals (e.g., body-wave codas), especially in regions with poor station coverage (e.g., big recording gaps). Compared with the well-recovered body-wave phases, the surface wave portion shows relatively large uncertainties in amplitude recovery (see Fig. 10e). The degraded reconstruction performance is mainly challenged by the waveform complexity of surface wave, which is characterized by a dispersive wave train rather than a distinct (linear) phase arrival (e.g., body waves). To alleviate these effects, a frequency-dependent, instead of a linear, time window may be required to better isolate the surface wave energy. For example, one may consider a Gaussian window with varying center frequencies as widely adopted in dispersion analysis[43]. As

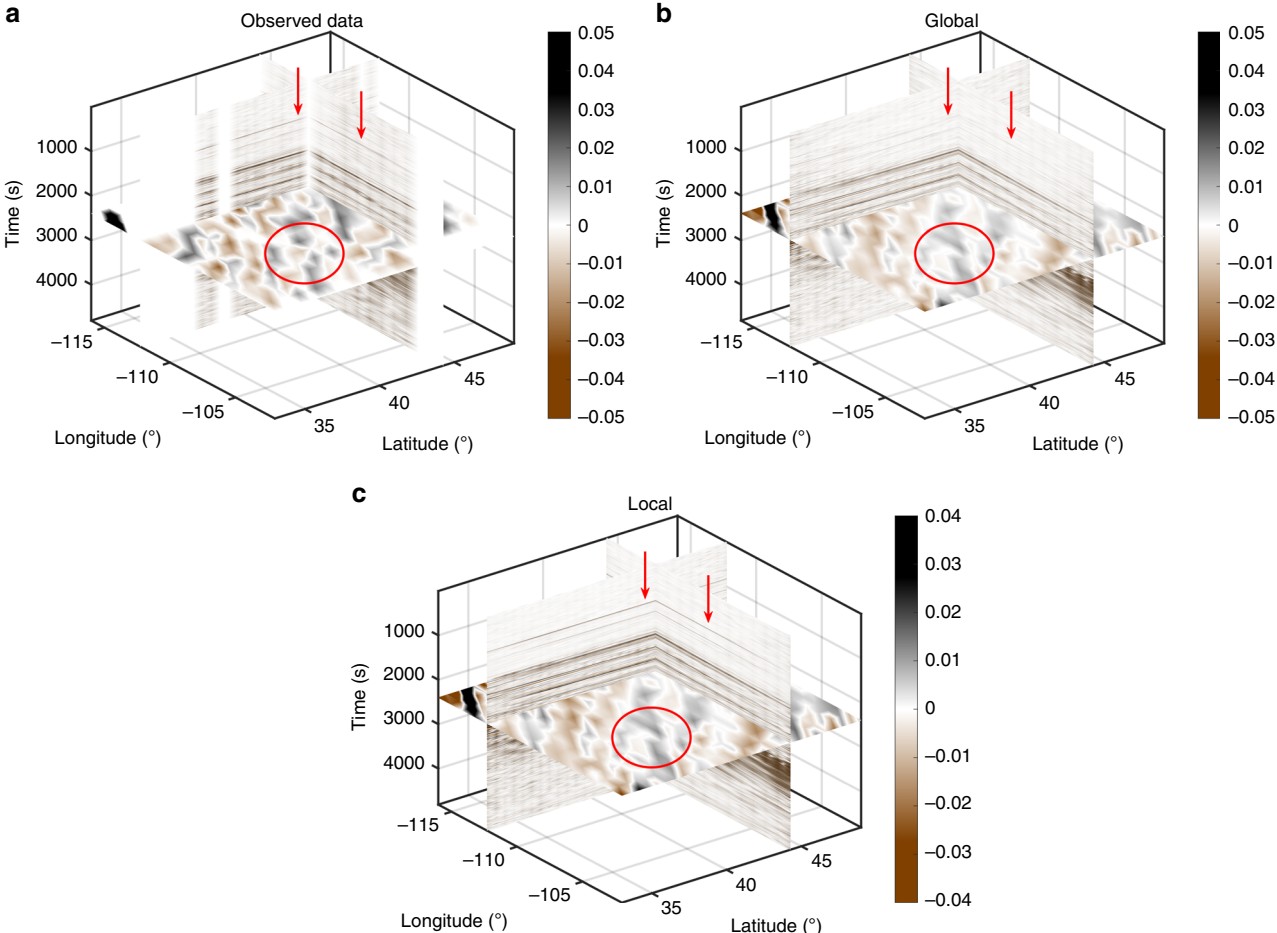

**Fig. 8** A 3D view of the earthquake data recorded by USArray analyzed in our study. **a** Observed data on a regular grid. The circled area shows the region with poor spatial sampling. The red arrows indicate the weak-amplitude P-wave arrivals contaminated by the noise. **b** Reconstructed data using the global rank-reduction method. **c** Reconstructed data using the localized rank-reduction method

importantly, a more careful examination (and selection) of surface wave related rank values is also essential to better capture the surface wave energy. Both aspects are critical to an improved reconstruction performance of surface waves and are worth future investigations. The lateral distribution of uncertainty is relatively constant across the study area except at the southwest corner (Supplementary Fig. 11), where uncertainty is about three times higher due to highly insufficient data sampling. Through waveform comparisons and uncertainty analysis, we are confident to suggest that the proposed localized rank-reduction method enables a robust reconstruction of the incomplete earthquake recordings, in particular, body-wave phases, which provides useful data constraints to the regions where no seismometers were placed.

## Discussion
The synthetic and real data examples demonstrate the ability of localized rank-reduction method in reconstructing the missing traces. The complete dataset, with improved quality and spatial sampling, is critical for improving array-based seismic imaging methods[12,13]. In this section, we demonstrate the accuracy of our method and its application in seismic imaging using the cross-correlation travel time measurement, a widely used technique in travel time tomography.

Seismic tomography is one of the most commonly applied seismic imaging techniques that greatly improve the understanding

of the internal structure of Earth. Seismic tomography can be broadly classified based on the types of data (travel time or waveform) and the approximation of wave propagation theory (ray or finite frequency)[44]. One classical method of the tomographic family is regional travel time tomography[45] that utilizes the travel time differences between nearby stations to resolve the subsurface velocity structure. The density of the station and the accuracy of travel time measurements are critical factors for high-resolution imaging. We provide an example of travel time measurements to demonstrate the capability of the proposed reconstruction method in improving P wavefield.

The relative travel times between stations in a recording array are measured using multi-channel cross-correlation[46]. For a pair of stations, the optimal relative travel time is determined by the time delay that leads to the maximum correlation coefficient between the two traces containing the phase of interest (e.g., P wave). To ensure the consistency of travel time measurements among all recording stations, the relative travel time between each station pair is optimized through a least-squares inversion. This optimization process also imposes a zero average constraint to the travel times. Finally, the demeaned theoretical arrival times (based on a reference Earth model) are subtracted from the optimized values to obtain the relative travel time residuals that can be inverted for velocity perturbations underlying the recording array.

The travel time measurement is first performed on the original data. We filter the seismic traces between 0.03 and 0.125 Hz to

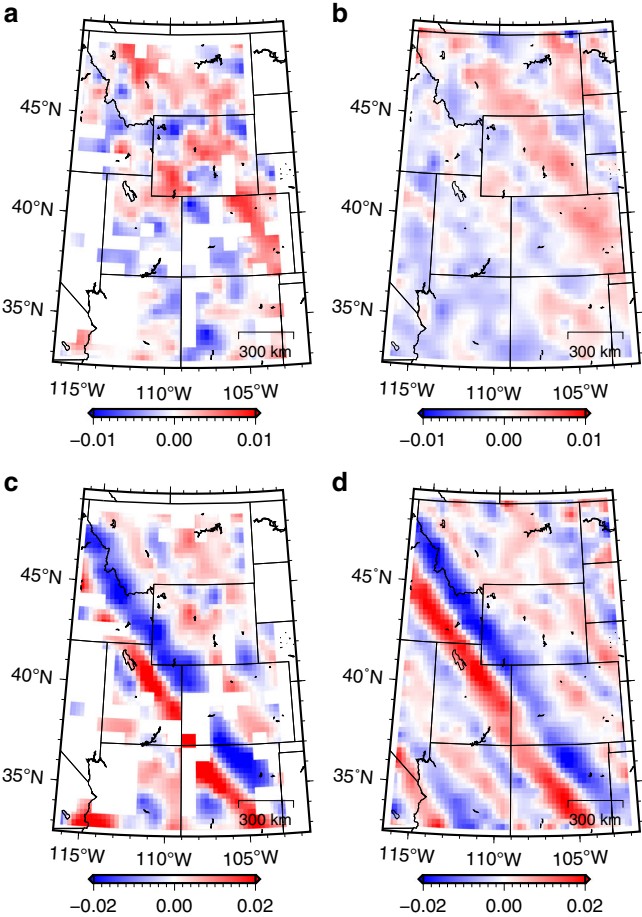

**Fig. 9** A comparison of reconstruction results between different time slices. **a** The time slice corresponding to $t = 1200$ s of observed USArray data. **b** The time slice corresponding to $t = 1200$ s of reconstructed USArray data. **c**, **d** The same as (**a**, **b**) but for time slice at 2100 s. The clear phase arrivals are reconstructed in the SW corner of the seismic array

measurements. For a plane wave from a teleseismic source, it illuminates the upper mantle portion of the Earth structure beneath the recording array from the direction of back-azimuth at a steep incidence angle. The presence of negative (i.e., slow) and positive (i.e., fast) velocity anomalies, respectively, causes the delay and advance of the P-wave arrivals. The availability of high-resolution seismic velocity models in the western US enables a detailed examination of the travel time accuracy. We utilize a recent high-resolution tomographic model of the upper mantle of continental US[9], which is constructed using 516,668 P-wave travel time residuals measured within multi-frequency bands in combination with a finite-frequency kernel[47]. In addition to the high model resolution, the choice of this model is prompted by the same travel time type (i.e., relative residuals) that permits a direct comparison with our measurements. Since travel time residual reflects an integrated effect of velocity anomalies in the upper mantle, in particular, the heterogeneous mantle lithosphere, we consider velocity structures between 50 and 250 km and compute the average amplitude (i.e., perturbation) of the seismic velocity anomaly (Fig. 11c). The resulting velocity perturbation agrees well with the reconstructed travel time pattern, whereas its correlation with the travel times from original data is less apparent. In particular, the high-velocity zone beneath the southern Wyoming craton is in excellent agreement with the coherent travel time advance in the reconstructed data, while the original data shows a much scattered travel time pattern. Furthermore, the sharp velocity transition to the low-velocity Cordillera is clearly delineated by the travel time variation from the reconstructed data, while this travel time contrast is largely smeared in the original data. In addition, some small-scale structural variations (e.g., the SE corner of the array and the Yellowstone anomaly) are captured by the travel times from the reconstructed data. More importantly, in the regions with large data gaps, the accuracy of the reconstructed travel times are supported by the velocities. For example, in the western part of the array between 36 and 43°N, the travel time shows an overall delay pattern that corresponds to below than average seismic velocities, and the smaller data gap at the NE corner also exhibits a reasonable agreement between the two variables.

We further evaluate the uncertainty of travel time measurements from both datasets. For each station, we compute its travel time errors with respect to other stations defined as the difference between the least-squares optimized and cross-correlation determined travel times. The standard deviation of all measurement errors is then used as an estimate of the measurement uncertainty at this station. The respective average uncertainties of the original (Fig. 12a) and the reconstructed data (Fig. 12b) are 0.50 s and 0.41 s, respectively. We notice that the uncertainty of the reconstructed data is in fact dominated by a few large outliers at poorly constrained nodes (see Fig. 12b), where the absolute value of travel time residual is on average >2.0 s. Most of these nodes are located in the regions with the largest data gap or near the edges of the data volume (e.g., SW corner). These anomalously large measurements are typically caused by cycle-skipping or poor data quality, and are removed from the inversion process in regional tomographic study[48]. After removing the outliers, which accounts for 4.5% of the total measurements, the average uncertainty decreases to 0.35 s and is significantly smaller than that of the original data, suggesting an improved P-wave consistency in the reconstructed data.

The travel time example demonstrates the improved resolving power of the complete dataset to small-scale structures. Thanks to the rapid development of dense seismic arrays around the globe, new seismic imaging techniques have been focusing on simultaneous processing of recordings from nearby sensors on a uniform/semi-uniform grid. One of the rapidly advancing fields is

enhance the useful signal and utilize a 100 s cross-correlation window starting 50 s prior to the predicted P-wave arrival time based on AK135 model[38]. The resulting travel time pattern is similar to the predictions (i.e., theoretical arrival times from AK135) because the time perturbations induced by structural variation is much smaller than the move-out (Supplementary Fig. 12). After correcting for the move-out effect, the remaining values (i.e., relative travel time residuals) are considered to be mainly caused by the receiver-side velocity anomalies. The measured residuals from the original data generally show a dichotomy of travel times: delay in the SW and advance in the NE (Fig. 11a), suggesting distinctive mantle structures in these two regions. Some local-scale variations are also observable in the travel times (e.g., time delay related to Yellowstone hotspot). We then conduct travel time measurements on the reconstructed data using identical parameters (i.e., frequency and window length). The resulting travel time variation is more coherent compared with that from the original data (Fig. 11b). A major difference is a significant time advance in the central part of the array, which transitions sharply to the regime of time delay in the SW. In comparison, the travel time advance in the original data appears to be more scattered and the transition is much smoother.

Seismic travel time is sensitive to velocity anomalies within the Fresnel zone around the ray path[47], thus the consistency of the travel time pattern with the velocity structures can be used as an effective metric to evaluate the robustness of the travel time

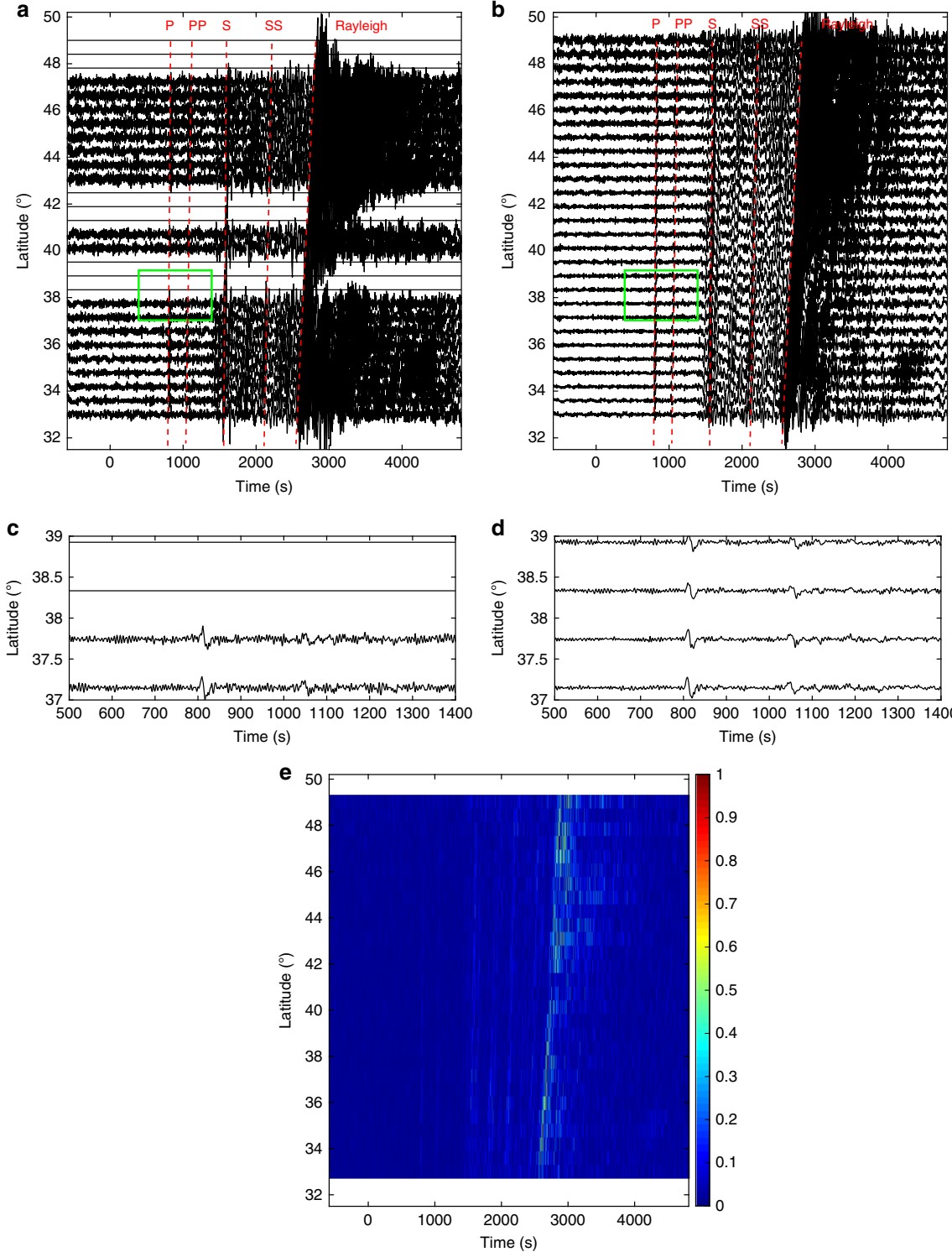

**Fig. 10** A detailed comparison of reconstruction performance along with a longitudinal slice (longitude = 107.6°W). **a** Raw waveforms before reconstruction. **b** Reconstructed waveforms. The main phase arrivals are marked by the red lines and labeled. The missing traces have been reconstructed and the signal-to-noise ratio of the main phases are significantly improved in (**b**). **c**, **d** Detailed comparison of the weak-amplitude phases (P and PP) that are highlighted by the green rectangles in (**a**, **b**). **e** The reconstruction uncertainty for the longitudinal slice

seismic surface tomography that utilizes either the travel time[4,49] or the shape of the wavefield[14,50] recorded at dense arrays to infer the elastic properties of the subsurface. These methods typically require the calculation of spatial derivatives with respect to a certain parameter such as the travel time (e.g., Eikonal tomography[4]) or amplitude of the surface wave (e.g., Helmholtz

tomography[14] and gradiometry[50]), which is best performed on a regular grid to ensure the numerical stability and accuracy. The compressive-sensing-based reconstruction approach proposed by Zhan et al.[36] has demonstrated the improvement in resolving velocities using the reconstructed wavefields. Similarly, we expect an improved imaging performance of these gradient-based

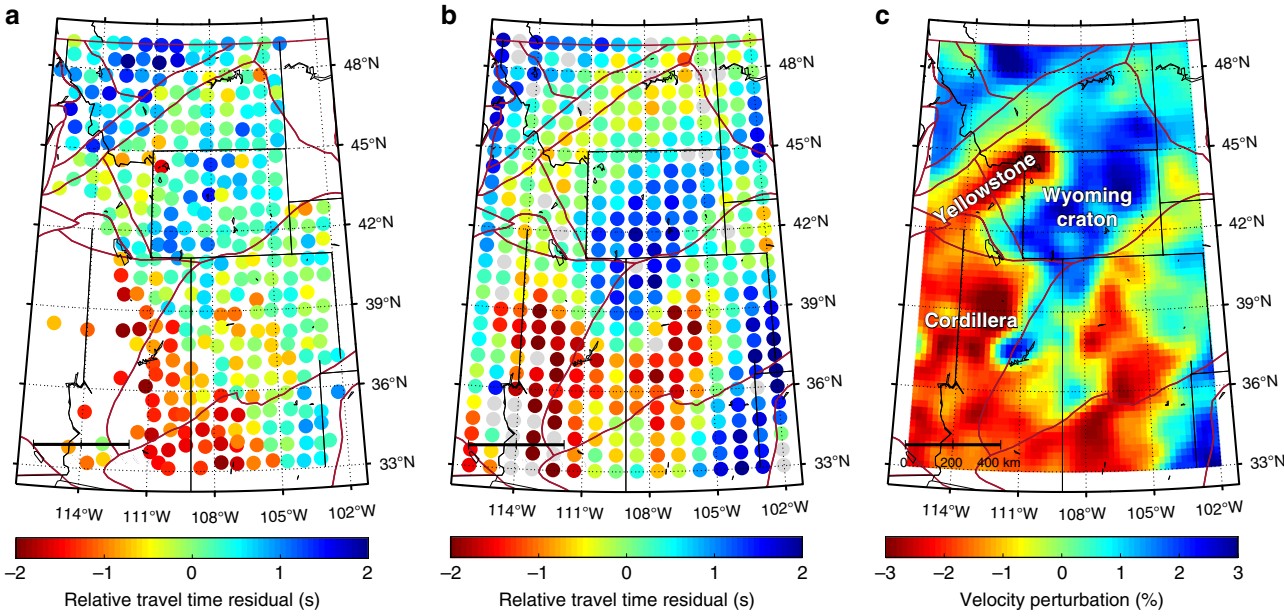

**Fig. 11** A comparison of P-wave travel time measurements between the original and reconstructed data. **a** Relative travel time residuals across the recording array measured from the original data. **b** Relative travel time residuals measured from the reconstructed data. Measurements with uncertainties larger than one standard deviation of the cross-correlation error are represented by gray circles in (**b**). The brown lines indicate the boundaries of major tectonic domains in the western US. **c** Average P-wave velocity perturbations between 50 and 250 km depths from Schmandt and Lin[9]

tomographic approaches when applied to the reconstructed data. The validation of this argument is beyond the scope of this paper and will be investigated in future studies. Finally, the improvement in sensor technology as exemplified in several large N array experiments[51–54] has permitted a higher-density sampling that is close to an exploration-scale survey. The development of imaging technology, such as the one proposed in our study, should work hand-in-hand with the improvement in data acquisition to achieve the ultimate goal of having a better understanding of the Earth's structure.

## Methods

**Iterative rank-reduction method**. The incomplete earthquake data on a regular grid can be expressed simply as

$$\mathbf{u} = \mathbf{S}\mathbf{v}, \tag{1}$$

where $\mathbf{v}$ denotes the complete earthquake data, $\mathbf{u}$ denotes the observed incomplete data, and $\mathbf{S}$ is the sampling operator. Both $\mathbf{u}$ and $\mathbf{v}$ denote vectors of size $N_t N_x N_y \times 1$. $N_t$, $N_x$, $N_y$ denote the lengths of the $t$, $x$, $y$ axes, respectively. The sampling operator is a diagonal matrix of size $N_t N_x N_y \times N_t N_x N_y$. In the sampling operator, each "zero" diagonal entry corresponds to a missing sample and "one" diagonal entry denotes a sampling point. The sampling matrix will be presented when analyzing the results of data reconstruction.

Given a reconstruction filter $\mathbf{P}$, we follow the weighted projection-onto-convex sets (POCS)-like scheme to iteratively reconstruct the missing earthquake data and suppress the strong random noise that downgrades the quality of the data. The weighted POCS-like method is expressed as

$$\mathbf{v}_n = \alpha_n \mathbf{u} + (1 - \alpha_n \mathbf{S})\mathbf{P}\mathbf{v}_{n-1}, \tag{2}$$

where $\alpha_n$ is an iteration-dependent relaxation factor to control the suppression of the random ambient noise. Here, we follow the linear-decreasing strategy used in Chen et al.[25] for iteratively suppressing the noise while restoring the noisy data with newly interpolated data.

In the rank-reduction method[23], the filtering is conducted in the frequency-space domain of the multi-channel earthquake data. When rewriting $\mathbf{v}$ in a tensor form $V(y, x, t)$, the rank-reduction method first transforms the data from $t-x-y$ domain to $f-x-y$ domain, i.e., $V(y, x, f)$. The frequency domain data are then rearranged into a multiple of block Hankel matrices for each frequency slice. The Hankel matrix $\mathbf{R}_i$ for row $i$ of the matrix $V(y, x, f)$ constructed from the frequency slice of $f$ is

$$\mathscr{R}_i(f) = \begin{pmatrix} V(i,1,f) & V(i,2,f) & \cdots & V(i,M_x,f) \\ V(i,2,f) & V(i,3,f) & \cdots & V(i,M_x+1,f) \\ \vdots & \vdots & \ddots & \vdots \\ V(i,L_x,f) & V(i,L_x+1,f) & \cdots & V(i,N_x,f) \end{pmatrix}. \tag{3}$$

By omitting $f$, Eq. (3) is then inserted into a block Hankel matrix by

$$\mathscr{H} = \begin{pmatrix} \mathscr{R}_1 & \mathscr{R}_2 & \cdots & \mathscr{R}_{M_y} \\ \mathscr{R}_2 & \mathscr{R}_3 & \cdots & \mathscr{R}_{M_y} \\ \vdots & \vdots & \ddots & \vdots \\ \mathscr{R}_{L_y} & \mathscr{R}_{L_y+1} & \cdots & \mathscr{R}_{N_y} \end{pmatrix}, \tag{4}$$

where the Hankel matrix $\mathscr{H}$ is assumed to be of low rank. $N_x$ and $N_y$ denote the number of channels in the $x$ and $y$ dimension of the data. $L_x = \lfloor N_x/2 \rfloor + 1$ and $M_x = N_x - L_x + 1$, where $\lfloor \cdot \rfloor$ is the operator to calculate the integer part. Similarly, $L_y = \lfloor N_y/2 \rfloor + 1$ and $M_y = N_y - L_y + 1$. The low-rank extraction of the key information from the Hankel matrix is equivalent to the principal component analysis of the Hankel matrix. To optimally extract the principal components, i.e., the signals, from the Hankel matrix of an updated data, we aim at solving the following optimization problem

$$\begin{aligned} \min \|\mathscr{E}\|_F^2 \\ \text{s.t. } \text{rank}(\mathscr{S}) = K, \text{and}, \mathscr{E} = \mathscr{H} - \mathscr{S}, \end{aligned} \tag{5}$$

where $\mathscr{S}$ denotes a low-rank matrix and $\mathscr{E}$ denotes small random perturbations. Problem 5 can be conveniently solved via the singular value decomposition (SVD) algorithm.

The SVD of $\mathscr{H}$ can be expressed as

$$\mathscr{H} = \mathscr{P}\Sigma\mathscr{Q}^T, \tag{6}$$

where $[\cdot]^T$ denotes transpose, and

$$\begin{aligned} \mathscr{P} &= [\mathbf{p}_1, \mathbf{p}_2, \cdots, \mathbf{p}_N], \\ \Sigma &= [\sigma_1, \sigma_2, \cdots, \sigma_N], \\ \mathscr{Q} &= [\mathbf{q}_1, \mathbf{q}_2, \cdots, \mathbf{q}_N]. \end{aligned} \tag{7}$$

$\mathbf{p}_i$ and $\mathbf{q}_i$ ($1 \le i \le N$) are $i$th left and right singular vectors of SVD. $\sigma_i$ denotes the $i$th singular value. $N$ denotes the number of columns in matrix $\mathscr{H}$. The decomposed value matrices are used to reconstruct a rank-reduced matrix by

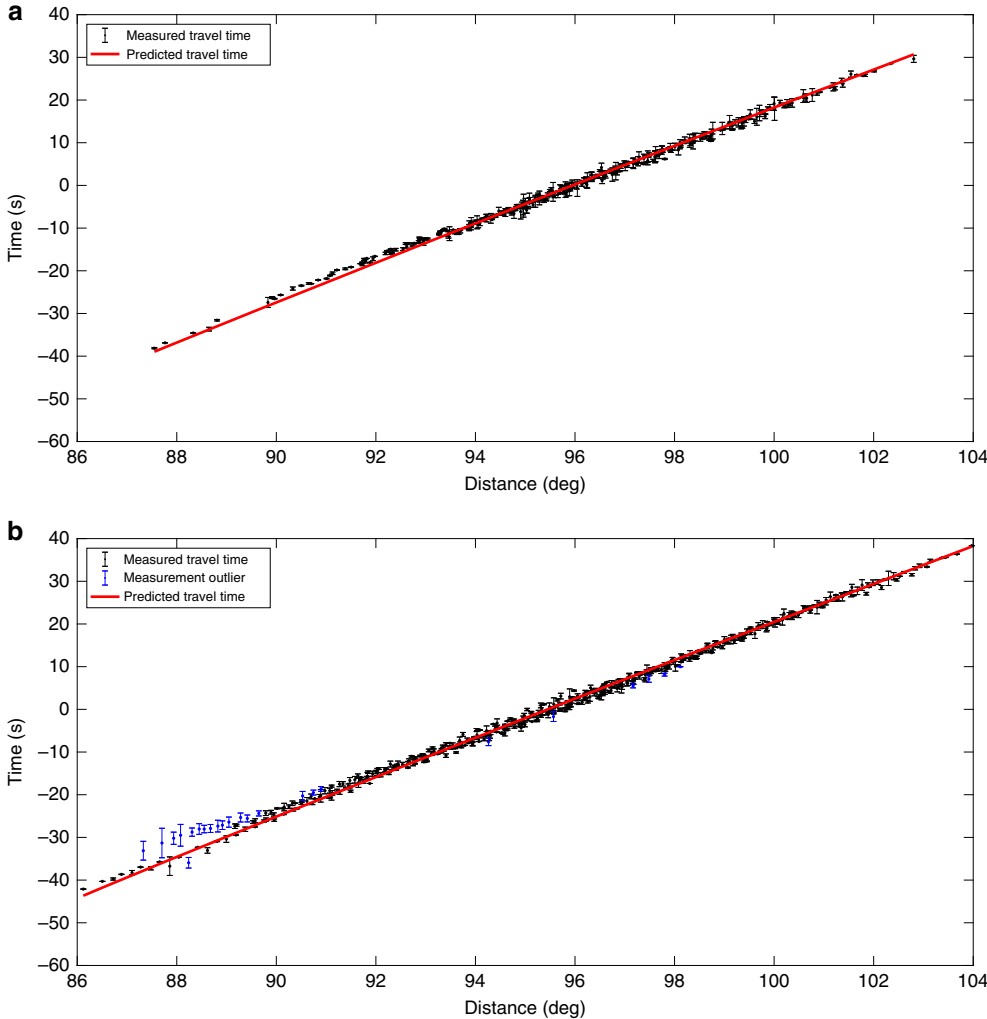

**Fig. 12** A comparison of travel time measurement uncertainties. **a** Travel time plot of the original data. **b** Travel time plot of the reconstructed data. Both measured and predicted travel times are demeaned to emphasize the perturbations. The blue data points in (**b**) represent measurement outliers differing by more than 2 s from the predictions, which are excluded from tomographic inversion in real practice

selecting only the first $K$ singular vectors and singular values

$$\hat{\mathscr{H}} = \sum_{i=1}^{K} \sigma_i \mathbf{p}_i \mathbf{q}_i^T . \tag{8}$$

The rank-reduced Hankel matrix is then remapped back to a 1D vector by averaging along the anti-diagonals. The mapping from each frequency slice to the Hankel matrix is referred to as the Hankelization step and the averaging along anti-diagonals is called the inverse Hankelization process. The overall process including the Hankelization, principal component extraction, and inverse Hankelization can be denoted by the restoration filtering operator **P** expressed previously in Eq. (2). Note that similar strategies are commonly used in the subspace separation methods, e.g., defining the log-likelihood of each eigenvalue to be related to signal subspace to estimate the signal component.

**Automatic rank selection**. In the rank-reduction filtering introduced above, the predefined rank parameter $K$ plays a significant role in obtaining satisfactory performance. It has been documented that an appropriate selection of $K$ can be the number of seismic events that have distinct slownesses[23]. However, the real earthquake data are never a simple composition of several wave types with clear arrivals. Instead, the earthquake data can be much more complicated, e.g., the seismic arrivals are buried in the strong noise or the weak phases are hidden in the more dominant strong codas. Besides, the events in the earthquake record can be curving when the range of epicentral distance is large.

Thus, the selection the optimal rank is a non-trivial task. Generally speaking, if the rank is chosen too large, the rank-reduction filter tends to preserve all subtle features in the data, including the random noise and missing seismograms, thus the resulting data will contain strong residual noise and not be able to reconstruct the true-amplitude of the missing data. If the rank is chosen too small, the rank-

reduction filter can remove most useful signal components and only leave the most dominant energy in the seismic record, e.g., the coda waves or the PP phases with strong amplitude. The rank parameter is highly correlated with the structural complexity of the data, and traditionally the optimal rank requires a lot of human efforts and prior knowledge.

Thus, we introduce an adaptive method to optimally select the rank parameter for the rank-reduction filtering. We utilize the difference of the singular values between signals and noise in the singular spectrum to define the rank. We first define a singular value ratio sequence

$$r_i = \frac{\sigma_i}{\sigma_{i+1}}, i = 1, 2, 3, \cdots, N - 1, \tag{9}$$

where $\sigma_i$ is the $i$th singular value of the Hankel matrix $\mathscr{H}$ and $\{r_i\}$ denotes the singular value ratio sequence. $N$ denotes the size of the singular spectrum. The optimal rank $K$ is obtained when the sequence $r_i$ reaches the maximum

$$\hat{K} = \arg \max_i r_i. \tag{10}$$

The principles of the adaptive rank selection method introduced in Eqs. (9) and (10) are based on the detection of the cutoff rank in the singular value spectrum which indicates the separation between signal and noise energy.

## Data availability

All broadband seismic waveforms are retrieved from IRIS-DMC (Incorporated Research Institutions for Seismology, Data Management Center) (http://ds.iris.edu/ds/nodes/dmc/). Results obtained in this study are available upon requested from the corresponding author.

## Code availability

The codes of 3D seismic reconstruction are available from the corresponding author upon reasonable request.

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

## Acknowledgements

The earthquake data used in this study is requested from Incorporated Research Institutions for Seismology (IRIS). The authors are grateful to Jinwei Fang, Shaohuan Zu, and

Mi Zhang for insightful discussions. The research is partially supported by the Thousand Youth Talents Plan, and the Starting Funds from Zhejiang University.

## Author contributions

Y.K.C. designed the project. Y.K.C. and M.B. processed the synthetic and field data. Y.K.C. and Y.F.C. analyzed the results. Y.K.C., M.B., and Y.F.C wrote the paper. All authors reviewed the paper.

## Competing interests

The authors declare no competing interests.
