## [Peer Review File · Nature Communications]

Reviewers' comments:

Reviewer #2 (Remarks to the Author):

Review of "Obtaining free USArray data by multi-dimensional seismic reconstruction by Chen et al." NCOMMS-19-10375

General overview

In this paper, authors present an application of an iterative rank-reduction method to simultaneously densify and denoise the array observations. The study includes a synthetic test and an additional application on USArray data. Results are interesting and the method might be of the interest of the community. However, I think the manuscript requires some revisions before being considered for publication. Please see the details of my comments and suggestions in the following.

Main comments

It would be nice if the authors can provide a zoomed view of some of the reconstructed traces in Figure 11 along with their neighboring traces to see any pieces of evidence of an interpolation-like phenomenon.

Overall uncertainties have been estimated only for the travel-time measurements. How about the reconstructed traces? during the rank reduction process basically some new data interties have been generated. How confident we can be about these reconstructions? Could you provide any measure of uncertainty estimation for the reconstructed traces?

In Figure 10b we see that the reconstructed image provides a higher resolution of an image (on the left) where the overall patterns are more or less along what we expect from the low-sampled image (left one). The only way that we can validate the newly reconstructed image is to cross-validate with areas where higher sampling reveals some pattern. However, in Fig10 it seems in areas like Montana and/or Wyoming, where existing data already cover most of these states, after the reconstruction the patterns have been changed. High-velocity region at the eastern side of Montana, for instance, is mostly represented with low-velocities in the original map. Could you please explain why this has happened? How can we assure that this higher resolution does not come with a cost of lower overall confidence?

I am not sure if "reconstruction" is the best term to use here. The method densifies the recording field by adding some sort of estimation for places where no data is originally available not fixing problematic existed data.

Figure 7 and 8 and the relevant description of binning can be moved to the supplementary materials.

The red circles in Figure 9 do not match the low-sampled areas of Figure 8 and more correspond to highly sampled areas. Moreover, it is hard to see the improvements in Fig9b and Fig9c while it seems the reconstruction process has changed the pattern in the south-west of the circle. What would be the reason for this? is there any chance that the method can add some artifacts?

Minor comments

Please see the annotated PDF for my comments.

Reviewer #3 (Remarks to the Author):

This paper presents a potentially powerful methodology based on local wavefield interpolation to reconstruct teleseismic wavefields from unevenly distributed array recordings, in the presence of noise.

The authors first present a synthetic test, followed by an application to the wavefield from a teleseismic event recorded on the USArray network while it was stationed in the western US.

While intriguing, I feel that the methodology presentation and demonstration lacks in several respects.

First, what strikes me is a lack of discussion of the frequency content and spatial wavelength content that surely is essential to the recovery of a more complete signal, given the available bandwidth, spatial distribution of stations and noise characteristics that may all contribute to aliasing of the reconstructed wavefield if not properly conducted.

Also, the synthetic test is, in my opinion, poorly designed and therefore not very convincing of the applicability of the method to real data. Why not build a realistic model of the same region as covered by the real data, with different spatial scales, including some beyond resolution with available spatial and temporal sampling, as well as realistic seismic noise, that can be obtained from real data (as opposed to arbitrary 10% white noise)?

Without that, there is no way of judging if the wavefield reconstructed from the real data is robust or not. Comparison with the tomographic image does not help, since the latter are at best smoothed versions of the real structure (in fact there is some lateral shift of the whole image which should at least be commented on).

My opinion is that the authors need to make a substantial effort to present their methodology in a better documented, convincing way, and I encourage them to do so.

The paper is also riddled with small grammatical errors, of which I was able to highlight a certain number while reading, but I'm sure there are more to be found.

Reply to the reviews of manuscript
”Obtaining free USArray data by multi-dimensional seismic reconstruction”
by Yangkang Chen, Min Bai, and Yunfeng Chen

We thank the three referees and the editor for constructive suggestions. We have taken into account all the recommendations and have changed the paper accordingly. The revision focuses on four main points:

1. We performed a new seismic wave simulation to produce the synthetic data. To make this simulation as realistic as possible, we 1) carefully designed a two-layer velocity/density model with Moho depth constraint obtained from a recent tomographic model (Schmandt et al., 2015), 2) synthesized the plane wavefield by solving the 3D elastic wave equation based on a finite-difference scheme on a GPU-embedded workstation, 3) utilized the stacked P wave of the actual earthquake recording as the effective source-time function for wave simulation, and 4) added realistic noise based on the noise level in the USArray data example. The mathematical details of our wave simulation approach is provided in the supplementary material.
2. We provided uncertainty estimates for the reconstructed seismic traces as suggested by Reviewers 2 and 3. We conducted a bootstrapping test (Efron and Tibshirani, 1991) to estimate the uncertainties in different areas of the reconstructed data. Specifically, we randomly selected 40% from the observed seismograms to reconstruct the whole 3D seismic cube and repeated this step for 20 times. The normalized standard deviation of these 20 trials is used as the uncertainty estimation for each reconstructed point in this 3D data cube. A new figure (Figure 10e) has been added to demonstrate the uncertainty
3. We thoroughly revised the method section and added significant clarification on our methods. We have presented the methodology in more detail, utilized more standard mathematical formulation, provided the full list of parameters, and most importantly, carefully explained how we adjust the parameters to obtain the best result.
4. We discussed in detail the influence of frequency and spatial wavelength contents to signal recovery. The testing results are presented as new supplementary figures (Figures 10-12).

Please also see the revised paper with marks for detailed modifications (e.g., the line numbers mentioned below). The original comments are in *Italics*, replies are in **Bold**, and call-outs to the revised manuscript are shown in blue.

REFEREE 1

1. *Please find my comments on the manuscript entitled ”Obtaining free USArray data by multi-dimensional seismic reconstruction” by Yangkang Chen et al., submitted to Nature Communications.*

In the present manuscript, the authors present a strategy for both interpolating and denoising seismic data recorded on dense seismic arrays at spatial location where no sensor is present. The method is based on a local rank-reduction method, and shows better results in comparison with the state-of-the art global rank-reduction method. The discussion is of quality, the conclusions are well supported by the data. I therefore consider the manuscript could be indeed considered for being published.

I have however some concerns about the manuscript and recommend moderate revisions before further processing. Indeed, there is a general lack of explanation in several

major points. For instance, the motivations of this work are never explained properly and the choice of parameters used to run the strategy are arbitrary chosen and sometimes never discussed. I have included below a detailed list of comment with other minor points, with the hope that it could help to strengthen the manuscript.

Reply: Thanks for your recognition. We have made significant changes to address all your concerns.

2. *Abstract & introduction: The authors do not clearly state why seismologists would benefit from having a regular array.*

Reply: Thank you for pointing this out. One important reason is that it is more convenient to analyze, process, and visualize the data in regular arrays, since most existing processing and imaging algorithms for seismic data are based on regular arrays. This has been clarified in the introduction (lines 20-21) and read as: "The high-quality data with a regular spatial sampling can greatly benefit data analysis, processing and visualization, and also improve the numerical stability and accuracy for grid-based seismic imaging techniques (e.g., Eikonal and Helmholtz tomography)."

3. *Synthetic data test: There is a lack of information about the dataset. What are the observed seismic phases? What information is represented in color, and what are the values of the color scale? Does this seismic data comes from a wave propagation solver, or is it obtained with a handmade analytic signal? If you designed it with analytic waveforms, why did you not considered a wave propagation solver, so we could observe the quality of your strategy with respect to a heterogeneous velocity model?*

Reply: Thanks for your suggestion. We have replaced the previous synthetic data with a new synthetic data simulated by solving a plane-wave propagation problem. To simulate plane waves, we evenly allocate point sources on a plane beneath the study area. We solve the elastic wave equation using a finite-difference method on a GPU-embedded workstation. The stacked P wave of the USArray data example is used as our source wavelet for wave simulation. We also design a realistic structural model based on the suggestion by Reviewer 3. This model honors the Moho depth variation in the study area based on a recent tomographic model (Schmandt et al., 2015). The elastic parameters (velocity and density) of the crust and upper mantle are obtained from the average values of a 1D reference Earth model (Kennett et al., 1995). We have replaced all analysis and discussions based on the new synthetic model. The old example is provided in the supplementary material.

4. *What is the bandwidth of the synthetic data? By eye, it looks quite narrow around a dominating frequency, which is an advantage when applying any denoising strategy. How does your strategy deals with noisy broadband data?*

Reply: We thank reviewer for pointing this out. We have carefully examined the frequency content of synthetic data and the dominating frequency of the synthetic data is confined below 0.3 Hz (supplementary Figure 9). The SSA method transforms the data into frequency-space domain and reconstruct the data for each frequency slide. So the method is essentially applied to a series of narrow-band signals. As long as the signal preserves spatial coherence in the frequency band of interest, the recon-

struction method can work well. This has been clarified on lines 142-155 in the revised manuscript.

5. *MSSA has never been defined (only SSA). Please, clarify.*

Reply: MSSA stands for multi-channel singular spectrum analysis. To avoid confusion and make the paper easy to follow, we consistently use the term rank-reduction method (same as SSA method) in the revised manuscript.

6. *What is the influence of the 3D moving window size? How did you choose to take a 100 x 10 x 10 window? How can you have a 50 point overlap in each dimension when some dimension are only 10?*

Reply: The 3D moving window makes the windowed data better meets the assumption of the low-rank reconstruction method, i.e., local plane-wave assumption. If the window is too small, the spatial correlation is weak and reconstruction performance be deteriorated. If the window is too large, the windowed data still contain curving seismic events. We have added new discussions (see lines 128-141) regarding how we determine the window size (and rank) in the revised manuscript. In the case of a 50-point overlap when one dimension is only 10, we need to extend the dimension to the window size (e.g., 100), which is obviously not reasonable since we extend too much for a large window size. In such case, we would prefer to use a window size of 10 points with a 5-points overlap. Regarding the optimal selection of window size, we have explained in the revised manuscript that **"The length of window in each dimension should be chosen so as to segment the data into patches with the smallest extension. For example, for a dimension of length 100, the recommended window sizes are 10, 20, 50 or 100, since other choices will all cause an extension of dimension."**

7. *How did you select a rank-reduction of 3? Is there any analytic reason for that? How does the results vary with different rank values?*

Reply: We use an automatic rank selection method as detailed in the paper. Here, 3 means a reference rank. The algorithm can adapt the rank parameter according to the input data. The rank is associated with the number of distinct dipping components. Since we use local windows, the data can be viewed as plane waves locally. We roughly assume that in each window we have 3 distinct dipping components, although not perfect, the automatic rank selection strategy can make it more reasonable.

8. *PCA and plane waves: did you consider other models of decomposition that are not limited to plane waves? Could you discuss this in the manuscript?*

Reply: Thank you for the suggestion. As far as we know, most of models of decomposition are based on the plane-wave assumption in the seismological research field. Non plane-wave based decomposition methods can offer potential solution to the limitation of current reconstruction method, and is a topic worth investigating further. One potential solution might be extending the current classic PCA theory to the non-stationary version (Zhao and Shang, 2016), where we can take the curvature of seismic waves into consideration. In our study, the non plane-wave effects are largely

alleviated by using the local window and the output windowed data can be approximated by plane waves. We have added more discussions on the validity and drawback of, as well as potential improvement to, the plane-wave assumption on lines 90-101 of the revised manuscript as:

"This method divides the data cube into several smaller volumes to minimize the curvature of seismic waves, which essentially imposes a local plane-wave constraint to alleviate the non-planar effect. More sophisticated methods based on non plane-wave assumption (e.g., non-stationary principal component analysis (Zhao and Shang, 2016)) may offer better solutions to the curved wavefronts."

9. "We also use local similarity³⁵, which measures the similarity between two signals in a local sense, to evaluate the reconstruction performance."

How is calculated the local similarity ? Please provide a relevant definition in the manuscript. What is the need of Figure 4?

Reply: We have provided a clear definition of the local similarity metric in supplementary material of the revised manuscript. Figure 5 provides a 3D rendering of local similarity matrix that is used to demonstrate where we have higher reconstruction accuracy. The values in Figure 5 also indicates the uncertainty of the reconstruction algorithm, e.g., lower local similarity indicates lower accuracy and higher uncertainty.

10. "In both cases, the parameters are fine-adjusted to ensure the best results."

What are (1) the full list of parameters and (2) the criterion that you use to claim that you "ensure" the best results? It is not correct to present a strategy without fully describing the way you implement it and test it. If you made a study for defining the best set of parameters, it must appear here.

Reply: Thanks for your suggestion. The parameters for both strategies are convenient to control. We have provided detailed explanations of the full list of parameters and how we tune the parameters. These are summarized in a new paragraph (lines 128-141) of the revised manuscript and read as: "The parameters for both strategies are fine-adjusted to achieve the best reconstruction results. The only parameter for the global rank-reduction method is the rank. To determine the optimal value, we linearly increase the rank and select the one that maximizes the SNR of the reconstructed data. On the other hand, a two-step process is adopted to determine a pair of parameters (i.e., rank and window size) for the localized rank-reduction method. As a first step, we optimize the window size while considering a relatively large reference rank (e.g., five). The exact choice of reference rank is not critical at this stage since the rank selection will be further optimized by the automatic rank selection method. We fix the overlapping between two neighbor windows to be half of the window size. The length of window in each dimension should be chosen so as to segment the data into patches with the smallest extension. For example, for a dimension of length 100, the recommended window sizes are 10, 20, 50 or 100, since other choices will all cause an extension of dimension. With all the possible combinations of the window sizes considering all three dimensions, we select the best window size leading to the largest SNR. In the second step, we further optimize the reference rank for the selected window size. We decrease the reference rank and adopt the new value if it can further improve the SNR. In real data processing, we use the same strategy except for the criterion to evaluate the output

performance, which is prohibited by a lack of ground truth solution (i.e., term s in the equation of SNR). Instead, we define the maximum cross-correlation value between a reconstructed missing trace and its nearest observed trace as the criterion.”

11. *Figure 3: Could you please provide some insight on why the global method induces a time shift?*

Reply: In this case, the global method behaves like a strong smoother. For a typical seismic waveform, when we apply a smoother onto it, it will cause a time shift between the wave peaks before and after the smoothing.

12. *Figure 4: What should we read on the colorbar? A label free colorbar does not make any sense.*

Reply: Thank you for pointing this out. Colorbar means local similarity. We have added the label for the colorbar.

13. *USArray data test: What is the need to integrate to displacement?*

Reply: We integrate the velocity to displacement field to directly assess the ground motion. The type of seismogram (velocity vs. displacement) would not affect the reconstruction results.

14. *”In global reconstruction, we treat the whole 3D data cube as the input and set the rank to eight, which is determined empirically by our tests.”*

Again, what is the criterion for selecting the ”best empirical tests”? You must provide a dedicated paragraph to this. Could also please mention the rank that you used in the local method?

Reply: We have provided detailed explanations on the selection process and criteria for the optimal parameters of both global and localized methods. In rank for localized method is first set to a constant reference rank (i.e., 3) and then this value is optimized during automatic rank selection process based on the characteristics of the data in each window. Please refer to the response to question 10 above for detail.

15. *”We then compare two time slices between the original and the reconstructed data.” Where? You never make any reference to Figure 10a.*

Reply: Sorry for the mistake. We have properly referred to Figure 10a in the revised manuscript.

16. *Iterative rank-reduction method: What are u , S and V depending on? Space, time, frequency? Are they matrices, vectors, of which dimension? Please, provide proper definitions.*

Reply: We have provided detailed definitions. \mathbf{v} denotes the complete earthquake data, \mathbf{u} denotes the observed incomplete data, and \mathbf{S} is the sampling operator. Both \mathbf{u} and \mathbf{v} denote vectors of size $N_t N_x N_y \times 1$. N_t, N_x, N_y denote the lengths of the t, x, y axes, respectively.

17. *The notations used for defining the problem in equation 5 and 6 are the ones used in programming, not in mathematics. You should reformulate with appropriate notations.*

Reply: Thank you for the suggestions. We have revised the notations.

18. *Automatic rank selection: You should clearly state in the main text that you actually use an automated rank selection strategy. As you state, it is a challenging task, and*

your method seems to perform good with it.

Reply: Thanks for your insight. We have revised following your suggestion, which can be found on lines 132-133 in the revised manuscript.

19. *Did you considered subspace separation methods such as AIC, which consider the log-likelihood of each eigenvalue to be related to signal subspace, and thus define the noise and signal subspaces?*

Reply: Thanks for your insight. We did not considered other methods and the performance of these methods are definitely worth investigating in future studies. In this revision, we have added discussions suggesting that similar strategies are commonly used in the subspace separation methods such as AIC, e.g., defining the log-likelihood of each eigenvalue to be related to signal subspace to estimate the signal component.

We thank Reviewer 1 for the detailed comments and insightful suggestions which significantly improved the impact of this study.

REFEREE 2

1. *In this paper, authors present an application of an iterative rank-reduction method to simultaneously densify and denoise the array observations. The study includes a synthetic test and an additional application on USArray data. Results are interesting and the method might be of the interest of the community. However, I think the manuscript requires some revisions before being considered for publication. Please see the details of my comments and suggestions in the following.*

Reply: Thanks for your recognition. We have made significant changes to the manuscript to address these concerns.

2. *It would be nice if the authors can provide a zoomed view of some of the reconstructed traces in Figure 11 along with their neighboring traces to see any pieces of evidence of an interpolation-like phenomenon.*

Reply: Thanks for your suggestion. We have provided a zoomed view of the reconstructed traces in Figure 10 (previous Figure 11) along with their neighboring traces. We also clarified in the text (lines 201-204) that the reconstructed waveforms indeed capture the coherent structure of the dataset rather than arising from excessive smoothing of neighboring traces, which read as: "A more detailed examination of weak amplitude phases shows that 1) the waveform characteristics (e.g., phase and amplitude) of the existing traces are well preserved by the reconstruction algorithm without excessive smoothing effect and 2) the reconstructed traces well capture the coherent energy of the data, showing similar waveform quality to the nearby (observed) traces (Figures 10c and 10d)."

3. *Overall uncertainties have been estimated only for the travel-time measurements. How about the reconstructed traces? during the rank-reduction process basically some new data interties have been generated. How confident we can be about these reconstructions? Could you provide any measure of uncertainty estimation for the reconstructed traces?*

Reply: Thanks for your suggestion. We have provided the uncertainty

estimation using the bootstrapping method and used the normalized standard deviation as a metric for uncertainty of the reconstruction. These discussions are added to lines 205-215 in the revised manuscript as:

”We conduct a bootstrapping test (Efron and Tibshirani, 1991) to estimate the effect of spatial sampling of wavefield on reconstruction. We randomly select 40% of the observed seismograms to reconstruct the 3D data cube and repeat this step 20 times. We calculate the standard deviation of the 20 reconstructed datasets and use the normalized deviation as an estimate of uncertainty. The reconstruction uncertainty is low for body-wave phases (e.g., P and S) even in presence of data gap with intermediate size (~ 200 km) (Figure 10e), whereas the uncertainty is slightly higher in time ranges with weak arrivals (e.g., body wave codas), especially in regions with poor spatial distribution of stations (e.g., big recording gaps). On the other hand, a relatively large uncertainty is observed in surface waves (see Figure 10e). This increased uncertainty is mainly caused by the waveform complexity of surface wave, which is characterized by a dispersive wave train rather than a distinct (linear) phase arrival (e.g., body waves). The lateral distribution of uncertainty is relatively constant across the study area except for at the southwest corner (supplementary Figure 12), where uncertainty is about 3 times higher due to highly insufficient data sampling.”

4. *In Figure 10b we see that the reconstructed image provides a higher resolution of an image (on the left) where the overall patterns are more or less along what we expect from the low-sampled image (left one). The only way that we can validate the newly reconstructed image is to cross-validate with areas where higher sampling reveals some pattern. However, in Figure 10 it seems in areas like Montana and/or Wyoming, where existing data already cover most of these states, after the reconstruction the patterns have been changed. High-velocity region at the eastern side of Montana, for instance, is mostly represented with low-velocities in the original map. Could you please explain why this has happened? How can we assure that this higher resolution does not come with a cost of lower overall confidence?*

Reply: Thanks for your concern. The reason is that in this time slice, which in fact shows the PP phase, the scattering energy (noise) in P wave codas is very strong around the Montana/Wyoming area, as can be revealed in the waveforms comparison (around 1200s). So the strong amplitude in the original data mainly reflects the scattering energy from P-wave coda rather than the coherent PP phase. In our reconstruction, we not only complete the missing data, but also improve the existing data by removing noise, which leads to much clearer wavefield pattern in the reconstructed data (see Figures 9a and 9b). We have clarified this point on lines 192-195 of the revised manuscript as: ”The first time slice shows the energy related to PP phase (Figure 9a), the free surface multiple of P wave. This weak phase is severely contaminated by the scattering energy associated with the P-wave codas. As a result, the wavefield pattern of PP is largely incoherent across the recording array, even in the center and northeastern portions where station coverage is high.”

5. *I am not sure if “reconstruction” is the best term to use here. The method densifies the recording field by adding some sort of estimation for places where no data is originally available not fixing problematic existed data.*

Reply: “Reconstruction” here only refers to a type of algorithm that has

been explored in the exploration seismic for a long time. Here we want to re-emphasize that in our proposed method, we not only recover the missing traces but also remove the noise and thereby enhancing the existing data.

6. *Figure 7 and 8 and the relevant description of binning can be moved to the supplementary materials.*

Reply: Thanks for your suggestion, we have moved them to the supplementary materials.

7. *The red circles in Figure 9 do not match the low-sampled areas of Figure 8 and more correspond to highly sampled areas. Moreover, it is hard to see the improvements in Fig9b and Fig9c while it seems the reconstruction process has changed the pattern in the south-west of the circle. What would be the reason for this? is there any chance that the method can add some artifacts?*

Reply: First, the red circles area (about lon:-106°- -110° and lat: 38°-43°) correspond to highly-sampled areas. We have changed a time slice in Figure 8, where the improvement from (b) to (c) is more obvious. Yes, the pattern in the south-west part of the circle is indeed changed, but according to the spatial coherency of the wavefield, we can conclude that the reconstructed data is more reasonable. Regarding the possibility of adding artifacts, yes, but the chance is very small. As long as the parameters are set properly, the method only recover hidden components or remove random noise.

8. *Minor comments: Please see the annotated PDF for my comments.*

Reply: We have followed all your suggestion and corrected the mistakes.

We thank Reviewer 2 for constructive comments which greatly improved the quality of this study.

REFEREE 3

1. *This paper presents a potentially powerful methodology based on local wavefield interpolation to reconstruct teleseismic wavefields from unevenly distributed array recordings, in the presence of noise. The authors first present a synthetic test, followed by an application to the wavefield from a teleseismic event recorded on the USArray network while it was stationed in the western US.*

Reply: Thanks for your recognition. We have made significant changes to address all your comments.

2. *While intriguing, I feel that the methodology presentation and demonstration lacks in several respects.*

Reply: Thanks for your suggestion. We have thoroughly revised the methodology section and improved the demonstration in a few aspects. The overall presentation is much improved now.

3. *First, what strikes me is a lack of discussion of the frequency content and spatial wavelength content that surely is essential to the recovery of a more complete signal, given the available bandwidth, spatial distribution of stations and noise characteristics that*

may all contribute to aliasing of the reconstructed wavefield if not properly conducted.

Reply: Thank you for pointing out this important issue. We have carefully consider the suggested factors and significantly improve the discussions in the following aspects.

First, we have conducted a new test to examine the influence of frequency content to the recovery performance based on a new simulated dataset (see replies to question 4 for details about simulation). In summary, we found that the frequency content of the signal plays a major in the reconstruction. The high-quality reconstruction results are obtained within the dominating frequency band of the raw data. Based on our investigation, we also provide suggestions to reconstruction work in real practice. The detailed discussions are provided on lines 142-155 in the revised manuscript as below: "To investigate the influence of frequency content to the reconstruction performance of the presented methodology, we extract different frequency contents of the incomplete synthetic data for the reconstruction. The bandwidth of the simulated seismic data is mainly confined between 0 and 0.3 Hz (supplementary Figure 9), similar to that of the observed teleseismic P wave. We linearly increase the frequency band from 0.05 to 0.4 Hz at an interval of 0.05 Hz and perform reconstruction at each frequency slide (supplementary Figure 10). The seismic signals are well recovered in the reconstructed data at frequencies up to 0.3 Hz, showing more coherent P and Ps arrivals. The reconstruction performance degrades at higher frequencies (>0.3 Hz) with a lower degree of recovery of the missing traces. The performance of reconstruction is mainly limited by the weak high-frequency seismic signals with energy that is close to (or below) the noise level (see supplementary Figures 10g and 10h). We further perform a quantitative assessment of the reconstruction quality by computing the SNR of the output signal at each frequency (supplementary Figure 11). Compared with the raw data, the reconstruction improves the SNRs by a factor of two at all frequencies except for the higher end (>0.3 Hz) of the frequency spectrum, where the SNRs fall below the level of input values. Overall, our test results suggest that the frequency content of the signal largely controls the performance of reconstruction, and high-quality results are achievable within the dominating frequency band of signal." As a result, a careful frequency analysis of the data is recommended prior to applying the reconstruction algorithm."

We also designed a bootstrapping test (Efron and Tibshirani, 1991) to evaluate the influence of spatial distribution of stations to reconstruction. The reconstruction uncertainties for varying degree of spatial sampling of the wavefield is given by the standard deviation from multiple trials of bootstrapping test. The bootstrapping test results are presented in two new figures (Figure 10e in the manuscript and supplementary Figure 12) and the discussions are provided in lines 205-215 of the revised manuscript as: "We conduct a bootstrapping test (Efron and Tibshirani, 1991) to estimate the effect of spatial sampling of wavefield on reconstruction. We randomly select 40% of the observed seismograms to reconstruct the 3D data cube and repeat this step 20 times. We calculate the standard deviation of the 20 reconstructed datasets and use the normalized deviation as an estimate of uncertainty. The reconstruction uncertainty is low for body-wave phases (e.g., P and S) even in presence of data gap with intermediate size (~ 200 km) (Figure 10e), whereas the uncertainty is slightly higher in

time ranges with weak arrivals (e.g., body wave codas), especially in regions with poor spatial distribution of stations (e.g., big recording gaps). On the other hand, a relatively large uncertainty is observed in surface waves (see Figure 10e). This increased uncertainty is mainly caused by the waveform complexity of surface wave, which is characterized by a dispersive wave train rather than a distinct (linear) phase arrival (e.g., body waves). The lateral distribution of uncertainty is relatively constant across the study area except for at the southwest corner (supplementary Figure 12), where uncertainty is about 3 times higher due to highly insufficient data sampling.”

We further consider the effect of noise characteristics on reconstruction performance. We added the relevant discussions in lines 73-74 of the revised manuscript as: ”To examine the influence of noise to reconstruction, we consider the realistic noise level of USArray data, which is then applied to the synthetic example. The reconstruction results show that the presented methodology can work well in the presence of real seismic noise.”

4. *Also, the synthetic test is, in my opinion, poorly designed and therefore not very convincing of the applicability of the method to real data. Why not build a realistic model of the same region as covered by the real data, with different spatial scales, including some beyond resolution with available spatial and temporal sampling, as well as realistic seismic noise, that can be obtained from real data (as opposed to arbitrary 10% white noise)?*

Reply: Thank you for the suggestion. We agree that the previous model is far from ideal. We carefully consider the suggested simulation schemes and performed a new seismic wave simulation to produce the synthetic data. To make this simulation as realistic as possible, we 1) carefully designed a two-layer velocity/density model with Moho depth constraint obtained from a recent tomographic model (Schmandt et al., 2015), 2) synthesized the plane wavefield by solving the 3D elastic wave equation based on a finite-difference scheme on a GPU-embedded workstation, 3) utilized the stacked P wave of the actual earthquake recording as the effective source-time function for wave simulation, and 4) added realistic noise based on the noise level in the USArray data example. The mathematical details of our wave simulation approach is provided in the supplementary material.

5. *Without that, there is no way of judging if the wavefield reconstructed from the real data is robust or not. Comparison with the tomographic image does not help, since the latter are at best smoothed versions of the real structure (in fact there is some lateral shift of the whole image which should at least be commented on).*

Reply: We agree with the reviewer that the reconstruction results are not properly justified in the earlier version. So we have carefully designed a realistic synthetic model per your suggestions. Based on this new model, the reconstruction uncertainty caused by frequency content and spatial sampling variation have been discussed in detail in the revised manuscript. Please see our replies to questions 3 and 4 for more details.

6. *My opinion is that the authors need to make a substantial effort to present their methodology in a better documented, convincing way, and I encourage them to do so.*

Reply: Thanks for your suggestion. We have made significant revisions following your suggestions. The two main aspects of revision are 1) adding a

realistic synthetic model; 2) adding significant explanations on the methodology we are presenting; 3) adding significant discussions on the parameter selection, influences due to many realistic factors.

7. *The paper is also riddled with small grammatical errors, of which I was able to highlight a certain number while reading, but I'm sure there are more to be found.*

Reply: We have thoroughly checked the manuscript and corrected all the grammatical errors.

We thank Reviewer 3 for insightful suggestions on simulation and uncertainty analysis which significantly improved the impact of this study.

REFERENCES

- Efron, B., and R. Tibshirani, 1991, Statistical data analysis in the computer age: *Science*, **253**, 390–395.
- Kennett, B. L., E. Engdahl, and R. Buland, 1995, Constraints on seismic velocities in the earth from traveltimes: *Geophysical Journal International*, **122**, 108–124.
- Schmandt, B., F.-C. Lin, and K. E. Karlstrom, 2015, Distinct crustal isostasy trends east and west of the rocky mountain front: *Geophysical Research Letters*, **42**, 10–290.
- Zhao, X., and P. Shang, 2016, Principal component analysis for non-stationary time series based on detrended cross-correlation analysis: *Nonlinear Dynamics*, **84**, 1033–1044.

Reviewers' comments:

Reviewer #1 (Remarks to the Author):

Dear editor and authors,

Please find my review on the revised manuscript entitled "Obtaining free USArray data by multi-dimensional seismic reconstruction" by Yangkang Chen et al., submitted to Nature Communications.

In this new version, the authors have carefully integrated changes related to my previous comments. In particular, I read that the motivations are now clearly stated and of interest for the community. The synthetic tests are better performed with realistic synthetic tests made from finite differences simulations. This is a strong improvement to the former semi-analytic experiment that was far from being realistic. The theoretical materials are now well presented.

The discussion and conclusions are of quality and well supported by the data. I therefore consider the manuscript could be indeed considered for being published as it is.

Sincerely yours.

Reviewer #2 (Remarks to the Author):

authors have addressed my comments and suggestions and improved the manuscript significantly.

Reviewer #3 (Remarks to the Author):

The authors have made a significant effort to clarify their methodology and address reviewers' questions by redesigning their synthetic test which is now much more realistic. I am satisfied with most of the answers and below have some comments on details - The one important comment is written in bold and marked by **.

comments on details

Abstract: "acquisition of earthquake data" (remove "the")

text:

line 10: USArray

that "had been migrating" -> "that migrated"

line 25 : also methods such as "slant stacklet" see work of Sergi Ventosa and collaborators

line 35: "seismic arrays"

line 50 " three-dimensional 3D full wavefield": it's not "full" it's band limited !!

line 55: remove reference to Figure 7 - or put the current Figure 7 as Figure 1 (Figures have to be numbered in the order they are referred to in the text)

line 56: "is missing" (remove "the")

line 63: should be: "characterized by large depth variations"

line 73: "proceeds"  "preceeds"

line 76: I don't understand why you define the signal to noise ratio by subtracting the pure signal from the contaminated signal. Why not put the noise itself in the denominator, since you are using a sample of real noise?

In the Synthetic Test section, you should specify the frequency band of the synthetic data (since you later discuss recovery in different frequency bands) - and also indicate how large an event you consider/and at what distance, since that determines the signal to noise ratio..

line 110: "quality factors" - avoid this term as it has a specific meaning in seismology (the inverse of attenuation).

line 154: "dominating"-> "dominant" (please do indicate what that dominant frequency band is!) - if you are using real noise data, you will have a peak of noise around 6-7s (microseismic noise). In order to recover signal up to 0.3 Hz you must have large enough signal amplitudes in the synthetic data - perhaps a comparison of your "data" and "noise" spectra would help see clearly what conditions you are working under for this test.

Figure 1 caption: line 2: remove extra "and"

line 193: "scattering energy " -> "scattered energy"

 **lines 210-212: I am surprised you brush aside the question of surface wave reconstruction. It seems your study is really focused on body waves - so it is not correct to say that you are reconstructing the entire wavefield - not only is it band=limited, but you don't reconstruct the surface waves accurately - this is not a criticism, I expect an adaptation of the method to surface waves would be possible, but is outside the scope of the paper - it is important to stress that both in the abstract and in the discussion.

line 240 "Predicated" -> "predicted"

line 288 "global" -> "globe"

line 299: "such as the proposed.." -> "such as the one proposed..."

Reply to the reviews of manuscript
"Obtaining free USArray data by multi-dimensional seismic reconstruction"
by Yangkang Chen, Min Bai, and Yunfeng Chen

We thank the three referees and the editor for constructive suggestions. We have taken into account all the recommendations and have changed the paper accordingly. The revision focuses on addressing the comments and concerns from the third reviewer. Please also see the revised paper with marks for detailed modifications (e.g., the line numbers mentioned below). The original comments are in *Italics*, replies are in **Bold**, and call-outs to the revised manuscript are shown in blue.

REFEREE 1

- In this new version, the authors have carefully integrated changes related to my previous comments. In particular, I read that the motivations are now clearly stated and of interest for the community. The synthetic tests are better performed with realistic synthetic tests made from finite differences simulations. This is a strong improvement to the former semi-analytic experiment that was far from being realistic. The theoretical materials are now well presented.*

The discussion and conclusions are of quality and well supported by the data. I therefore consider the manuscript could be indeed considered for being published as it is.

Reply: Thanks for your recognition.

REFEREE 2

- Authors have addressed my comments and suggestions and improved the manuscript significantly.*

Reply: Thanks for your recognition.

REFEREE 3

- The authors have made a significant effort to clarify their methodology and address reviewers' questions by redesigning their synthetic test which is now much more realistic. I am satisfied with most of the answers and below have some comments on details - The one important comment is written in bold and marked by **.*

Reply: Thanks for your recognition and the final comments. We have revised the paper to address these remaining concerns.

- Abstract: "acquisition of earthquake data" (remove "the")*

Reply: Corrected.

- text: line 10: USArray that "had been migrating" → "that migrated"*

Reply: Corrected.

- line 25 : also methods such as "slant stacklet" see work of Sergi Ventosa and collaborators*

Reply: Thank you for pointing out this reference. We have properly cited this study in the revised manuscript.

5. *line 35: "seismic arrays"*
Reply: Corrected.
6. *line 50 " three-dimensional 3D full wavefield": it's not "full" it's band limited !!*
Reply: We have removed "full" from the discussion and clarified, wherever possible, that our reconstruction method targets on the body-wave phases.
7. *line 55: remove reference to Figure 7 - or put the current Figure 7 as Figure 1 (Figures have to be numbered in the order they are referred to in the text)*
Reply: Reference to Figure 7 has been deleted.
8. *line 56: "is missing" (remove "the")*
Reply: Corrected.
9. *line 63: should be: "characterized by large depth variations"*
Reply: Corrected.
10. *line 73: "proceeds" → "precedes"*
Reply: Corrected.
11. *line 76: I don't understand why you define the signal to noise ratio by subtracting the pure signal from the contaminated signal. Why not put the noise itself in the denominator, since you are using a sample of real noise?*
Reply: Thanks for your concern. The definition of SNR used in our study follows the conventions from signal/image processing and reflection seismology, which in fact represents the difference between the approximated and true solutions. In case of contaminated signal, the reviewer is right that the noise can be directly used in the denominator because the noise itself represents the deviation of contaminated signal from the true solution (i.e., noise free data). For the reconstructed data, the denominator no longer represents real noise in the raw data, which instead defines the deviation of the estimated signal from the clean signal and represents the reconstruction error (which we interpret as residual noise). Therefore, the adopted convention of denominator aims to provide a more general expression for assessing the uncertainty (and quality) of both input and output signals. We have clarified the definition of SNR on lines 81-83 of revised manuscript and read as: "This criterion is also applied to assess the quality of reconstruction, where \hat{s} represents the recovered data using the global or local reconstruction algorithms. Thus, the SNR measures the deviation of an estimated data from its true solution."
12. *In the Synthetic Test section, you should specify the frequency band of the synthetic data (since you later discuss recovery in different frequency bands) - and also indicate how large an event you consider/and at what distance, since that determines the signal to noise ratio.*
Reply: Thanks for your suggestion, we have specified the frequency band of the synthetic data. The frequency band has been clarified in the Synthetic test (lines 139-140) and read as: "high-quality results are achievable within the dominant frequency band (0-0.3 Hz) of the signal (Supplementary Figure 9)." As far as the magnitude and distance of the earthquake, we did not explicitly

define the epicenter distance and magnitude. Instead, these two parameters are implicitly accounted for in our simulation design. Specifically, we consider the actual earthquake location to calculate the incidence angle and direction of the plane wave. The actual magnitude (as well as the propagation effect) of earthquake is honored by taking the stacked P wave of real data. We have modified our descriptions to clarify these points in lines 68-72: “The physical parameters of an earthquake (e.g., epicenter and magnitude) are implicitly considered in our simulation. The direction of wavefront is determined from the epicenter distance and azimuth of the earthquake investigated in this study. The stacked P-wave of actual earthquake recordings is employed as the effective source-time function to honor the actual earthquake source parameters and ensure a similar frequency content and signal energy of the synthetics. The resulting synthetic data shows dominant frequencies between 0-0.3 Hz.”

13. *line 110: "quality factors" - avoid this term as it has a specific meaning in seismology (the inverse of attenuation).*

Reply: We thank Reviewer for raising this issue. To avoid confusion, we have changed “quality factors” to “quality metrics”.

14. *line 154: "dominating" → "dominant" (please do indicate what that dominant frequency band is!) - if you are using real noise data, you will have a peak of noise around 6-7s (microseismic noise). In order to recover signal up to 0.3 Hz you must have large enough signal amplitudes in the synthetic data - perhaps a comparison of your "data" and "noise" spectra would help see clearly what conditions you are working under for this test.*

Reply: We have corrected the typo and have specified the frequency band as 0-0.3 Hz. Besides, we have added the frequency spectra of noise and noise contaminated data in Figure 9 of the Supplementary material. The noises are mainly concentrated between 0.2-0.4 Hz and interfere with the P-wave energy.

15. *Figure 1 caption: line 2: remove extra "and"*

Reply: Corrected.

16. *line 193: "scattering energy" → "scattered energy"*

Reply: Corrected.

17. ***lines 210-212: I am surprised you brush aside the question of surface wave reconstruction. It seems your study is really focused on body waves - so it is not correct to say that you are reconstructing the entire wavefield - not only is it band=limited, but you don't reconstruct the surface waves accurately - this is not a criticism, I expect an adaptation of the method to surface waves would be possible, but is outside the scope of the paper - it is important to stress that both in the abstract and in the discussion.*

Reply: We thank reviewer for raising this concern. To address it, we have 1) clarified the frequency range and seismic phases that are tackled by the reconstruction method, and 2) expanded the discussions on the limitation and feasibility of the reconstruction of surface waves. These discussions can be found on lines 197-204 in the revised manuscript as: “Compared to the well-recovered body wave phases, the surface wave portion shows relatively large uncertainties in amplitude recovery (see Figure 10(e)). The degraded reconstruction

performance is mainly challenged by the waveform complexity of surface wave, which is characterized by a dispersive wave train rather than a distinct (linear) phase arrival (e.g., body waves). To alleviate these effects, a frequency-dependent, instead of a linear, time window may be required to better isolate the surface wave energy. For example, one may consider a Gaussian window with varying center frequencies as widely adopted in dispersion analysis [Dziewonski et al., 1969]. As importantly, a more careful examination (and selection) of surface wave related rank values is also essential to better capture the surface wave energy. Both aspects are critical to an improved reconstruction performance of surface waves and are worth future investigations.”

18. *line 240 "Predicated" → "predicted"*

Reply: Corrected.

19. *line 288 "global" → "globe"*

Reply: Corrected.

20. *line 299: "such as the proposed.." → "such as the one proposed..."*

Reply: Corrected.

Reply: We thank Reviewer 3 for additional comments that help clarify and improve the presentation of the manuscript.

REVIEWERS' COMMENTS:

Reviewer #3 (Remarks to the Author):

The authors have responded satisfactorily to my questions and comments